# CONDITIONAL GUIDED DIFFUSION PROBABILISTIC MODELS FOR IMAGE SUPER-RESOLUTION

## ABSTRACT

We propose a novel Conditional Guided Diffusion Probabilistic Model (CG-DPM) for image super-resolution. CG-DPM adopts diffusion models, which have strong abilities to generate diverse and photo-realistic images, through a stochastic iterative denoising process. The abilities can tackle the existing issue of over-smoothing artifacts in super-resolution tasks. The earlier work SR3 firstly uses diffusion models to conditional image generation for super-resolution. However, it simply upsamples the low-resolution images to the target resolution using bicubic interpolation as the conditional input, which cannot maximize the information of conditional images. In contrast, our CG-DPM involves conditional images in each different-scale level in the encoder so that the model can use the conditional images more effectively. We also introduce a separate score-based likelihood model to guide the original diffusion model to obtain a score-based posterior model. Moreover, since there is no analytic probabilistic formula to represent the likelihood probability for image super-resolution, we propose a novel scored-based loss function to train a separate guided network so that it can approximate the score-based likelihood probability. We conduct experiments on image super-resolution tasks for human faces and natural images at different scaling factors. CG-DPM achieves strong performance compared with existing methods. Meanwhile, the proposed method can also be used on other tasks, and more experiments show that our method achieves competitive results on the medical image segmentation.

## 1 INTRODUCTION

The single-image super-resolution aims to restore the high-resolution images from the corresponding low-resolution images, which contain unknown degradations. Many existing super-resolution methods Dai et al. (2019); Dong et al. (2014; 2015) learn a mapping from low-resolution images to high-resolution images with a pixel-wise constraint. These methods often result in perceptually unconvincing images with severe over-smoothing artifacts, though they can achieve remarkable results in terms of PSNR. To yield more photo-realistic results, GANs Ledig et al. (2017); Sajjadi et al. (2017); Wang et al. (2018) are employed. However, unnatural artifacts can still be observed in the generated images. Recently, diffusion models have been proposed and attracted numerous attention since diffusion models have the ability to generate diverse and photo-realistic images, which can tackle the existing issues of unnatural artifacts in super-resolution tasks.

Diffusion-based (*i.e*, DDPM Ho et al. (2020)) and scored-based (*i.e*, NCSN Song et al. (2020)) generative models, which we call both models "diffusion models" for brevity, have been proposed with similar ideas underneath but with two kinds of different perspectives. Firstly, the *diffusion process* utilizes $T$ steps of a small amount of isotropic Gaussian noise with gradually incremental standard deviations to corrupt the data $\mathbf{x}_0 \sim q(\mathbf{x}_0)$. Then, when $T$ is sufficiently large $T \to \infty$, $\mathbf{x}_T$ is equivalent to an isotropic Gaussian distribution. Meanwhile, diffusion models are trained to learn how to denoise each different step. Finally, diffusion models can construct desired data samples via a Markov chain that progressively denoises from Gaussian noise into a high-quality image. The Markov process of diffusion models is either based on Langevin dynamics algorithm Song et al. (2020) or learned via reversing the above *diffusion process* for score-based or diffusion-based generative models Sohl-Dickstein et al. (2015), respectively.

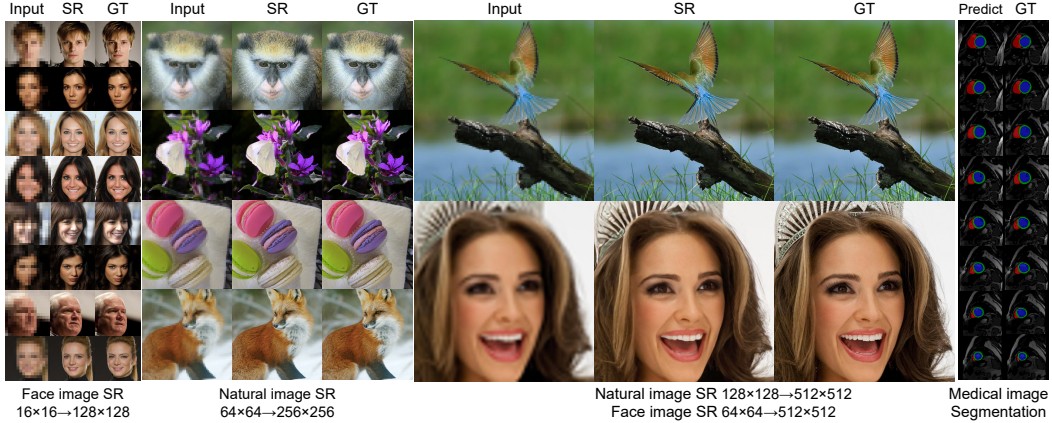

Input SR GT  Input  SR  GT  Input  SR  GT  Predict  GT

Face image SR
16×16→128×128

Natural image SR
64×64→256×256

Natural image SR 128×128→512×512
Face image SR 64×64→512×512

Medical image
Segmentation

Figure 1: The visualized predicted results of face ($16 \times 16 \rightarrow 128 \times 128$ and $64 \times 64 \rightarrow 512 \times 512$) and natural ($64 \times 64 \rightarrow 256 \times 256$ and $128 \times 128 \rightarrow 512 \times 512$) image super-resolution and medical image segmentation via our proposed CG-DPM.

Both diffusion models of DDPM Ho et al. (2020) and NCSN Song et al. (2020) have achieved excellent performance on unconditional image generative tasks. Afterward, many works Saharia et al. (2021); Tashiro et al. (2021); Song et al. (2020); Dhariwal & Nichol (2021) that adapt DDPMs to conditional image generation have made progress. The existing conditional diffusion models are mainly divided into two categories.

Training a single diffusion model involves conditional labels. For example, SR3 Saharia et al. (2022) simply upsamples the low-resolution image to the target resolution using bicubic interpolation and then concatenates with a noisy image $\mathbf{x}_t$ at $t$ step as input, which achieves strong performance. But the simple upsampling and concatenation cannot maximize the information of conditional images. Therefore, in this paper, we design the network that involves conditional images in each different-scale level in the encoder. In this way, our network can learn different scale information of conditional images. Meanwhile, our network can better learn how to denoise from Gaussian noise to real target data points by involving conditional images several times. But our redesigned model does arrive at the maximum capacity in this category, since the performance would always be affected via the stochastic iterative denoising process, which is the property of diffusion models, for pixel-level tasks (*i.e*, image super-resolution). One feasible solution is to introduce a separate score-based likelihood model, which is the second category introduced in the following part, to guide the stochastic iterative denoising process to real target data points closer to pursuing higher performance.

Training a separate score-based (the gradient of the log probability density function) likelihood model guides the original diffusion model to obtain a score-based posterior model. For example, Guided-Diffusion Dhariwal & Nichol (2021) achieves state-of-the-art performance on ImageNet via involving a guided classifier to achieve conditional diffusion models on class labels. Inspired by the guided network, we introduce a novel separate score-based likelihood model to guide an original diffusion model to obtain a score-based posterior model. Because there is no exact probabilistic formula to represent the likelihood probability for image super-resolution, we propose a novel scored-based loss function to train a separate guided network so that it can approximate the score-based likelihood probability. In this way, we can guide the stochastic iterative denoising process to real target data points closer.

Especially, for the redesigned method in the first category, we utilize high-resolution images as training data to implement *diffusion process* that progressively adds a small amount of Gaussian noise and set low-resolution images as conditional images to train conditional diffusion models. We conduct several experiments and find it is a benefit to involve low-resolution images in each different scale level in the encoder of the U-Net. For the second category, in order to pursue better performance, we train a separate model to approximate the likelihood score function via a score-based loss function (which we will call the separate model "guided network" for brevity). The input of the guided network consists of $\mathbf{x}_t$ that samples starting from white noise and denoises via a pre-trained diffusion model, and low-resolution images $\mathbf{y}$. Since $T$ can not be set to $\infty$ but usually thousands, there is a gap between $\mathbf{x}_T$ and an isotropic Gaussian distribution that we start sampling from. Meanwhile, our goal equips our model with the ability to sample from any random Gaussian

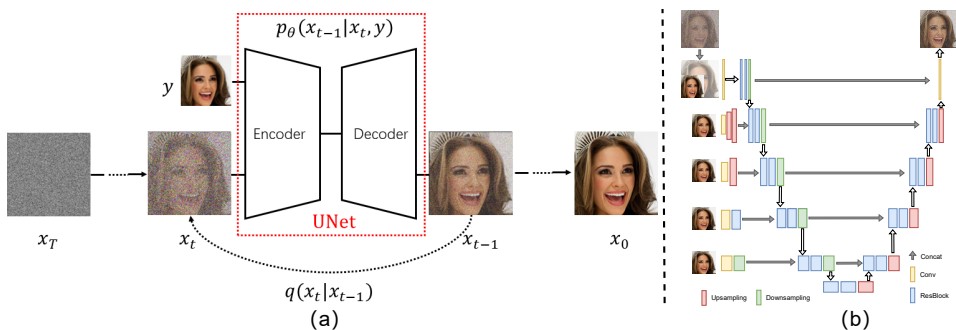

Figure 2: The overview of our proposed single conditional diffusion model. (a) performs the whole process of diffusion models. We utilize high-resolution images to implement the *diffusion process* that gradually adds a small amount of Gaussian noise and utilize low-resolution images as conditional images. (b) performs our proposed U-Net architecture, which involves different scales of conditional images into the encoder via a or several stacked up-sampling or down-sampling layers. The U-Net is trained to learn the *reverse process*.

noise to a certain data point in terms of a given low-resolution image. Given a pre-trained diffusion model, it can sample data points nearly a real target data point, which has a gap as well. Therefore, we build a guided network to learn the two gaps via the input images of $\mathbf{x}_t$ and $\mathbf{y}$ to learn how to represent the likelihood function.

To summarize, we make the following contributions: **(i)** We propose a novel conditional guided diffusion probabilistic model, named CG-DPM, for image super-resolution, and to achieve strong performance. **(ii)** In order for diffusion models to learn the information of conditional images adequately, we redesign the U-Net architecture that involves conditional images into each different scale level in the encoder. **(iii)** We train a separate guided network via a proposal and novel scored-based loss function to predict the scored-based likelihood function, which can guide our model to achieve higher performance. **(iv)** We conduct extensive experiments on image super-resolution tasks for human faces and natural images. Both qualitative and quantitative results demonstrate that our proposed CG-DPM achieves strong results compared with existing methods. Meanwhile, the proposed method can be used to solve other tasks, such as medical image segmentation.

## 2 RELATED WORK

**Diffusion Models.** Given a real data distribution $\mathbf{x}_0 \sim q(\mathbf{x}_0)$, we define the *diffusion process* that gradually adds a small amount of isotropic Gaussian noise with a variance schedule $\beta_1, ..., \beta_T \in (0, 1)$ to produce a sequence of latent $\mathbf{x}_1, ..., \mathbf{x}_T$, which is fixed to a Markov chain. When $T$ is sufficiently large $T \rightarrow \infty$ and a well-behaved schedule of $\beta_t$, $\mathbf{x}_T$ is equivalent to an isotropic Gaussian distribution.

$$q(\mathbf{x}_t|\mathbf{x}_{t-1}) = \mathcal{N}(\mathbf{x}_t; \sqrt{1 - \beta_t}\mathbf{x}_{t-1}, \beta_t\mathbf{I}), \tag{1}$$

$$q(\mathbf{x}_{1:T}|\mathbf{x}_0) = \prod_{t=1}^{T} q(\mathbf{x}_t|\mathbf{x}_{t-1}). \tag{2}$$

A notable Ho et al. (2020) property of the *diffusion process* admits us to sample $\mathbf{x}_t$ at an arbitrary timestep $t$ via directly conditioned on the input $\mathbf{x}_0$. Let $\alpha_t = 1 - \beta_t$ and $\bar{\alpha}_t = \prod_{i=1}^{T} \alpha_i$:

$$q(\mathbf{x}_t|\mathbf{x}_0) = \mathcal{N}(\mathbf{x}_t; \sqrt{\bar{\alpha}_t}\mathbf{x}_0, (1 - \bar{\alpha}_t)\mathbf{I}). \tag{3}$$

Since $q(\mathbf{x}_{t-1}|\mathbf{x}_t)$ depends on the data distribution $q(\mathbf{x}_0)$, which is intractable. Therefore, we need to parameterize a neural network to approximate it:

$$p_\theta(\mathbf{x}_{t-1}|\mathbf{x}_t) = \mathcal{N}(\mathbf{x}_{t-1}; \boldsymbol{\mu}_\theta(\mathbf{x}_t, t), \boldsymbol{\Sigma}_\theta(\mathbf{x}_t, t)). \tag{4}$$

We utilize the variational lower bound to optimize the negative log-likelihood.

$$\mathcal{L}_{\text{VLB}} = \mathbb{E}_{q(\mathbf{x}_{0:T})}\left[\log \frac{q(\mathbf{x}_{1:T}|\mathbf{x}_0)}{p_\theta(\mathbf{x}_{0:T})}\right] \geq -\mathbb{E}_{q(\mathbf{x}_0)} \log p_\theta(\mathbf{x}_0). \tag{5}$$

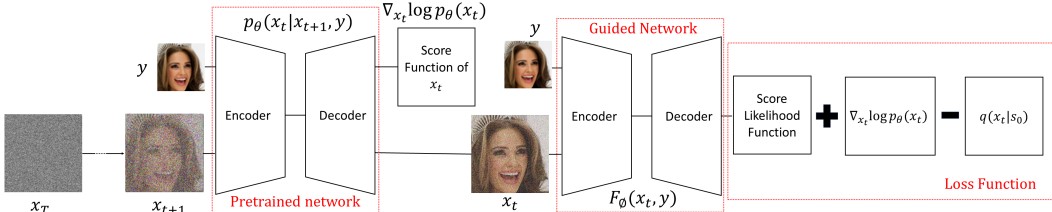

Figure 3: The overview of our proposed guided network. We utilize a pre-trained conditional diffusion model to sample $T - t$ steps to obtain $\mathbf{x}_t$ starting from Gaussian noise. We also utilize the pre-trained model to calculate the score function of $\nabla_{\mathbf{x}_t} log p_\theta(\mathbf{x}_t)$ by given $\mathbf{x}_t$ and $t$. Given a new training high-resolution image $s_0$, we can calculate $q(\mathbf{x}_t|s_0)$ by the *diffusion process*. We build a new network to approximate the score function of likelihood with $\mathbf{x}_t$ and conditional images $\mathbf{y}$ as the input. The loss function is expressed by the output of the trainable network, $q(\mathbf{x}_t|s_0)$ and $\nabla_{\mathbf{x}_t} log p_\theta(\mathbf{x}_t)$.

The objective function of the variational lower bound can be further rewritten to be a combination of several KL-divergence and entropy terms (see more details in Sohl-Dickstein et al. (2015)).

$$\mathcal{L}_{\text{VLB}} = \mathbb{E}_q[\underbrace{D_{\text{KL}}(q(\mathbf{x}_T|\mathbf{x}_0) \parallel p_\theta(\mathbf{x}_T))}_{L_T} \underbrace{- \log p_\theta(\mathbf{x}_0|\mathbf{x}_1)}_{L_0} + \sum_{t=2}^{T} \underbrace{D_{\text{KL}}(q(\mathbf{x}_{t-1}|\mathbf{x}_t, \mathbf{x}_0) \parallel p_\theta(\mathbf{x}_{t-1}|\mathbf{x}_t))}_{L_{t-1}}]$$

$L_0$ uses a separate discrete decoder derived from $\mathcal{N}(\mathbf{x}_0; \boldsymbol{\mu}_\theta(\mathbf{x}_1, 1), \boldsymbol{\Sigma}_\theta(\mathbf{x}_1, 1))$. $\mathcal{L}_T$ does not depend on $\theta$, it is close to zero if $q(\mathbf{x}_T|\mathbf{x}_0) \approx \mathcal{N}(0, I)$. The remain term $\mathcal{L}_{t-1}$ is a KL-divergence to directly compare $p_\theta(\mathbf{x}_{t-1}|\mathbf{x}_t)$ to *diffusion process* posterior that is tractable when $\mathbf{x}_0$ is conditioned,

$$\tilde{\beta}_t := \frac{1 - \bar{\alpha}_{t-1}}{1 - \bar{\alpha}_t} \cdot \beta_t, \qquad \tilde{\boldsymbol{\mu}}_t(\mathbf{x}_t\mathbf{x}_0) := \frac{\sqrt{\alpha_t}(1 - \bar{\alpha}_{t-1})}{1 - \bar{\alpha}_t}\mathbf{x}_t + \frac{\sqrt{\bar{\alpha}_{t-1}}\beta_t}{1 - \bar{\alpha}_t}\mathbf{x}_0, \qquad (6)$$

$$q(\mathbf{x}_{t-1}|\mathbf{x}_t, \mathbf{x}_0) := \mathcal{N}(\mathbf{x}_{t-1}; \tilde{\boldsymbol{\mu}}(\mathbf{x}_t, \mathbf{x}_0), \tilde{\beta}_t\mathbf{I}). \qquad (7)$$

**Conditional Diffusion Models.** Inspired by the successes of GANs in conditional image synthesis Mirza & Osindero (2014); Brock et al. (2018); Isola et al. (2017), it is reasonable to explore diffusion models with conditional labels. There are two main approaches to conditional diffusion models. Given a dataset denoted $\mathcal{D} = \{\mathbf{x}^i, \mathbf{y}^i\}_{i=1}^N$, $\mathbf{x}^i$ represents target images and $\mathbf{y}^i$ represents conditional labels. One approach of conditional diffusion models is to involve with conditional labels when training the network so that it can learn conditional transition distribution $p_\theta(\mathbf{x}_{t-1}|\mathbf{x}_t, \mathbf{y})$. Based on the idea, Saharia et al. (2021), and Tashiro et al. (2021) involve cleaned conditional images $\mathbf{y}$ into diffusion models and score-based models, respectively. Meanwhile, Song et al. (2020) replaces the cleaned conditional images $\mathbf{y}$ with diffused conditional images $\mathbf{y}_t$ that implement the same operation of *diffusion process* of $\mathbf{x}_t$. The other approach is to exploit a separate model to approximate the likelihood $p(\mathbf{y}|\mathbf{x}_t)$ and combine a pre-trained diffusion model to obtain the posterior. Sohl-Dickstein et al. (2015); Song et al. (2020) achieve it using the score function of the likelihood $\nabla_{\mathbf{x}_t} log p_\phi(\mathbf{y}|\mathbf{x}_t)$ to guide the *reverse process* to an arbitrary label $\mathbf{y}$. Based on the approach, Dhariwal & Nichol (2021) trains a classifier to guide the *reverse process* to an arbitrary class label on ImageNet. Diffusion models are utilized to solve inverse problems in image editing Song & Ermon (2019); Song et al. (2020); Kawar et al. (2022) and medical imaging Jalal et al. (2021).

**Image Super-Resolution.** Image super-resolution task is one domain of image-to-image translation tasks Isola et al. (2017); Zhang et al. (2016); Goodfellow et al. (2020). Many existing super-resolution methods Dai et al. (2019); Dong et al. (2014; 2015) learn a mapping from low-resolution images to high-resolution images with a pixel-wise constraint. These methods often result in perceptually unconvincing images with severe over-smoothing artifacts, though they can achieve remarkable results in terms of PSNR. To yield more photo-realistic results, GANs Ledig et al. (2017); Sajjadi et al. (2017); Wang et al. (2018) are employed. However, unnatural artifacts can still be observed in the generated images. Benefiting from the strong generative abilities of diffusion models that can generate high-quality and diverse images, SR3 Saharia et al. (2022) uses diffusion models to iteratively refine the noise, starting with pure Gaussian noise with low-resolution images as the conditional images. The model can generate high photo-realistic outputs due to the ability of diffusion models.

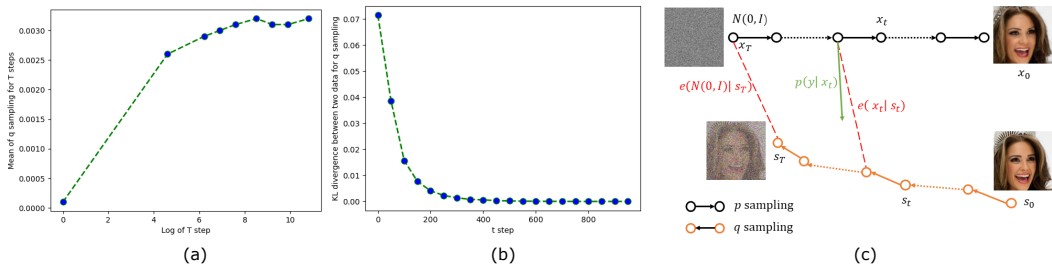

Figure 4: (a) illustrates the mean of $\mathbf{x}_T$ for the different $T$ step of the *diffusion process*. (b) illustrates the KL-divergence of two cases in different $t$ steps. (c) illustrates the straightforward idea of the guided network. $s_0$ is the target image. $\mathbf{x}_T$ to $\mathbf{x}_0$ is generated via the *reverse process*.

## 3 THE PROPOSED METHOD

Our proposed conditional guided diffusion probabilistic model, named CG-DPM, consists of two parts. One part is that we involve conditional images and redesign the architecture of the U-Net for the original diffusion model (Section 3.1). The other part is that we train a separate guided network to predict the score function of the likelihood $\nabla_{\mathbf{x}_t} log p_\phi(\mathbf{y}|\mathbf{x}_t)$ with a proposal score-based loss function (Section 3.2).

### 3.1 SINGLE CONDITIONAL DIFFUSION MODEL WITH INVOLVING CONDITIONAL IMAGES

Figure 2(a) illustrates the whole process for our single conditional diffusion model. We utilize high-resolution images as training data to implement *diffusion process* that progressively adds $T$ steps of a small amount of isotropic Gaussian noise with gradually incremental standard deviations. We redesign the U-Net in order to involve conditional images so that it can learn the reversing *diffusion process* via a parametric $p_\theta(\mathbf{x}_{t-1}|\mathbf{x}_t, \mathbf{y})$. Figure 2(b) illustrates our redesigned U-Net architecture.

We design several architectures and conduct experiments. We find that involving conditional images in each different scale level in the encoder achieves the best results. In our redesigned U-Net, we first use bicubic interpolation to up-sampling low-resolution images to the same size of high-resolution and then concatenate $\mathbf{x}_t$ as the input to the first level of the U-Net that consists of a symmetric encoder-decoder with skip connections that combine shallow, low-level, and fine-grained feature maps to deep, semantic, and coarse-grained feature maps. And then, we utilize one or several stacked Up-Sampling or Down-Sampling layers to do up-sampling or down-sampling for conditional images into different scale feature maps. These different scale feature maps will be concatenated with the very beginning of each level in the encoder. In this way, the U-Net can learn different scale feature maps of the conditional images, which can maximize the exploitation of the conditional images. In the reversing *diffusion process*, our model can be easy to arrive at the target point by involving the conditional images with many times and many scales.

Although we try lots of redesigned architectures, the performance does arrive at its maximum capacity. Since the attribute of the stochastic process for diffusion models is unfriendly for pixel-level tasks, the extended pixels cannot generate accurately. Although training a single diffusion model involving conditional images can bring a baseline performance, it can not bring a breakthrough in pixel-level tasks in this way. Therefore, we introduce a separate score-based guided network to guide the baseline model to arrive at higher performance in the next Section 3.2.

### 3.2 SCORED-BASED GUIDED NETWORK

Figure 3 illustrates the score-based guided network. Given a high-resolution image $s_0$ and a low-resolution image $\mathbf{y}$ as a pair of training data. Firstly, we obtain a pre-trained network from the above Section 3.1, start from white noise, and denoise $T - t$ steps to obtain $\mathbf{x}_t$. Secondly, we utilize $\mathbf{x}_t$ and $t$ to calculate the score function of $\mathbf{x}_t$. Thirdly, we calculate $q(\mathbf{x}_t|\mathbf{s}_0)$. Fourthly, we utilize $\mathbf{x}_t$ and the conditional image $\mathbf{y}$ as the input for the guided network $F_\phi(\mathbf{x}_t, \mathbf{y})$, which has the same architecture in Figure 2(b). And the output of the guided network has the same size of $\mathbf{x}_t$. Finally, we utilize the output of the guided network, score function of $\mathbf{x}_t$ and $q(\mathbf{x}_t|\mathbf{s}_0)$ to calculate the loss function to train the guided network.

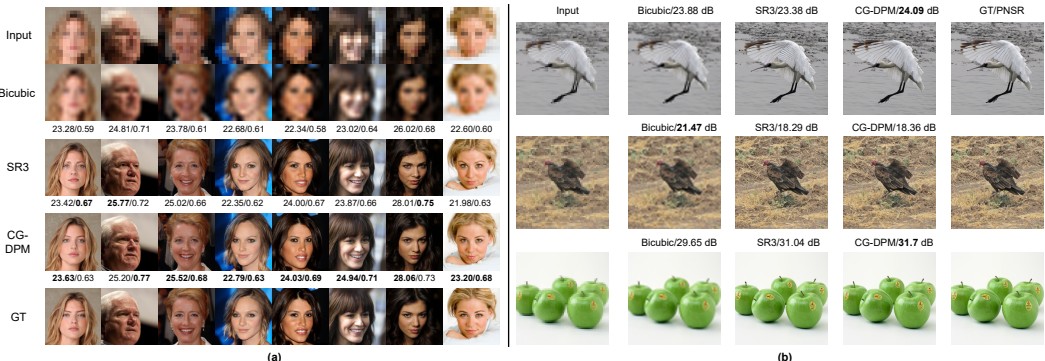

Figure 5: (a) Qualitative comparisons ($16 \times 16 \rightarrow 128 \times 128$), trained on FFHQ and evaluated on CelebA-HQ. (b) Qualitative comparisons ($64 \times 64 \rightarrow 256 \times 256$), trained on ImageNet and evaluated on ImageNet dev set. The indicators are PSNR (dB)/SSIM

**Selected T.** Ideally, $T$ should be sufficiently large $T \rightarrow \infty$ so that $\mathbf{x}_T$ is equivalent to an isotropic Gaussian distribution. In empirical, we usually set $T$ as 1,000 for the *diffusion process*. For the reversing *diffusion process*, we start from a Gaussian noise $\mathbf{x}_T \sim \mathcal{N}(0, I)$. Therefore, there is a question of whether the gap between $\mathbf{x}_T$ via 1,000 steps of the *diffusion process* and Gaussian noise is enough small so that we can utilize a Gaussian noise as the start point $\mathbf{x}_T$.

Therefore, we conduct experiments that select different $T$ (*i.e.*, 100, 500, 1,000, 2,000, 5,000, 10,000, 20,000, 50,000) and utilize $q(\mathbf{x}_T|\mathbf{x}_0)$ sampling for each $T$. For each $\mathbf{x}_T$, we calculate the mean and standard variance. Meanwhile, we also sample a Gaussian noise and calculate the mean is equal to 0.0001, and the standard variance is equal to 1. We find the standard variance of all cases is close to 1. But the mean of all $T$ steps is larger than 25 times. Figure 4(a) illustrates the results. Although the mean of all $T$ steps is close to 0.0030 that seems very small, the $\epsilon_\theta(\mathbf{x}_t, t)$ and the score function $\nabla_{\mathbf{x}_t} log p_\phi(\mathbf{x}_T)$ are close to 0.0001. Therefore, there is a huge gap between $\mathbf{x}_T$ from $q$ sampling and a Gaussian noise $\mathbf{x}_T \sim \mathcal{N}(0, I)$. Figure 4(c) illustrates the gap $e(\mathcal{N}(0, I)|\mathbf{s}_T)$.

**Disparity between Two Cases in Different Steps.** existing methods usually utilize traditional probability density functions to represent the likelihood function $p(\mathbf{y}|\mathbf{x}_t)$, such as the softmax function, and then train the network via traditional loss function, such as the cross-entropy loss function.

In image super-resolution tasks, it is hard to find an existing function to represent the likelihood function. Even if we can find a function to represent the likelihood function, it needs to meet the requirements that have low curvature compared to $\Sigma^{-1}$ and everywhere is non-zero. Our purpose is to obtain a target data point by giving a conditional image. Therefore, there is still a problem that the likelihood function should have the ability to distinguish the difference among any $\mathbf{x}_t$ that $t$ is around $T$. However, these $\mathbf{x}_t$ are extremely hard to distinguish since these are similar to Gaussian noise. Figure 4(b) illustrates the KL divergence of two cases in different $t$ steps. We can see that KL divergence is extremely small for more than 500 steps, which means it is very hard to distinguish two cases after 500 steps of $q$ sampling that progressively adds a small amount of Gaussian noise if you just directly distinguish based on two $\mathbf{x}_t$. Figure 4(c) illustrates the gap $e(\mathbf{x}_t|\mathbf{s}_t)$.

**Score-based Loss Function.** For image super-resolution tasks, it is extremely hard to represent the possibility density function of the likelihood $p(\mathbf{y}|\mathbf{x}_t)$. We tried using many functions that calculate similarity and distance to represent the likelihood and using a neural network to approximate. And all experiments got worse results since the non-scored-based functions destroy the structure of the original diffusion models. The approximated function must meet the requirements that have low curvature compared to $\Sigma^{-1}$ and everywhere is non-zero. Since the $\Sigma^{-1}$ of the *reverse process* has very low curvature, it must destroy the *reverse process* if the likelihood function has a high curvature.

Inspired by Chao et al. (2022), we approximate the likelihood function via a scored-based loss function. By the observation in Figure 4(c), the "green arrow" represents the score-based likelihood function that can be expressed via $e(\mathbf{x}_t|\mathbf{s}_t) + e(\mathcal{N}(0, I)|\mathbf{s}_t)$ ("red dash line") and $\nabla_{\mathbf{x}_t} log p_\phi(\mathbf{x}_t)$ ("black arrow"), which all have a relationship with $\mathbf{x}_t$ and conditional images $\mathbf{y}$. Therefore, we utilize $\mathbf{x}_t$ and $\mathbf{y}$ as the input images to extract useful information. We define a function to approximate the likelihood as below:

$$F_\phi(\mathbf{x}_t, \mathbf{y}) \approx \nabla_{\mathbf{x}_t} \log p(\mathbf{y}|\mathbf{x}_t). \tag{8}$$

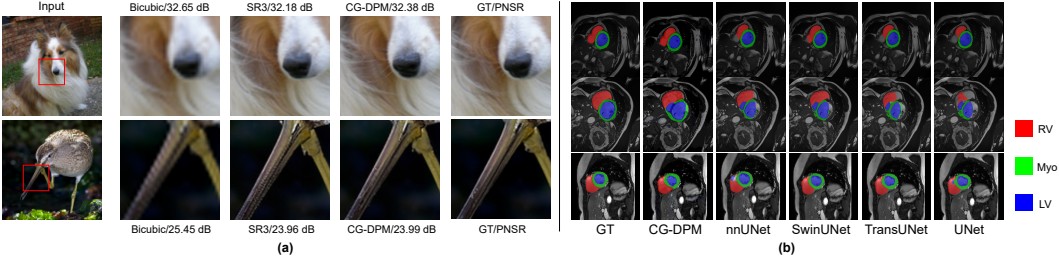

Figure 6: (a) Qualitative comparisons ($128{\times}128{\rightarrow}512{\times}512$), trained on ImageNet and evaluated on ImageNet dev set. (b) Qualitative comparisons on the ACDC dataset for medical segmentation.

We define the explicit score-based loss function as below and train a neural network to minimize the loss function,

$$\mathcal{L}(\phi) = \mathbb{E}_{p(\mathbf{x}_t, y)}\Big[ \parallel F_\phi(\mathbf{x}_t, \mathbf{y}) - \nabla_{\mathbf{x}_t}\log p(\mathbf{y}|\mathbf{x}_t) \parallel^2 \Big] \tag{9}$$

However, the loss term involves the true score-based likelihood function, which is intractable. We follow Bayes' theorem and the attributes of score functions, and formulate the below objective function as below (more details in the Appendix), $\mathcal{L}(\phi) =$

$$\mathbb{E}_{p(\mathbf{x}_{0:t}, y)}\Big[ \parallel F_\phi(\mathbf{x}_t, \mathbf{y}) + \nabla_{\mathbf{x}_t}\log p_\theta(\mathbf{x}_t) - \nabla_{\mathbf{x}_t}\log p(\mathbf{x}_t|\mathbf{s}_0) \parallel^2 \Big]. \tag{10}$$

Since we have a pre-trained model $\epsilon_\theta(\mathbf{x}_t)$ that predicts the noise added via the *reverse process*. And then, we can derive the score function:

$$\nabla_{\mathbf{x}_t} log p_\theta(\mathbf{x}_t) = -\frac{1}{\sqrt{1-\bar{\alpha}_t}}\epsilon_\theta(\mathbf{x}_t). \tag{11}$$

Based on the Equation 3, the last term of Equation 10 can be expressed as:

$$\nabla_{\mathbf{x}_t}\log p(\mathbf{x}_t|\mathbf{s}_0) = -\frac{\mathbf{x}_t - \sqrt{\bar{\alpha}_t}\mathbf{s}_0}{1-\bar{\alpha}_t}. \tag{12}$$

where $\mathbf{x}_t$ is sampled $T - t$ steps starting from $\mathbf{x}_T$ via the *reverse process* and $\mathbf{s}_0$ is a new target high-resolution image sampling from $p(\mathbf{x}_0)$. If we utilize the *diffusion process* to sample $t$ steps for the $\mathbf{s}_0$ and obtain $\mathbf{s}_t$. And we modify a little bit for Equation 12,

$$\nabla_{\mathbf{x}_t} log p(\mathbf{x}_t|\mathbf{s}_0) = -\underbrace{\frac{\mathbf{x}_t - \mathbf{s}_t}{1-\bar{\alpha}_t}}_{a} - \underbrace{\frac{\mathbf{s}_t - \sqrt{\bar{\alpha}_t}\mathbf{s}_0}{1-\bar{\alpha}_t}}_{b} = -\underbrace{\frac{\mathbf{x}_t - \mathbf{s}_t}{1-\bar{\alpha}_t}}_{a} - \underbrace{\frac{\sqrt{\bar{\alpha}_t}\mathbf{s}_0 + \sqrt{1-\bar{\alpha}_t}z - \sqrt{\bar{\alpha}_t}\mathbf{s}_0}{1-\bar{\alpha}_t}}_{b}.$$

where the term $a$ can be seen as the difference between $\mathbf{x}_t$ sampling via the pre-trained *reverse process* starting from a Gaussian and $\mathbf{s}_t$ sampling via the *diffusion process* at $t$ step. Since $z \sim \mathcal{N}(0, I)$, the term $b$ can be expressed as a Gaussian noise $\mathcal{N}(0, \frac{1}{1-\bar{\alpha}_t}I)$.

**Sampling.** Following the classifier sampling algorithms proposed in Dhariwal & Nichol (2021), we can modify and sample each reverse denoising step by two algorithms. The first sampling algorithm calculates $\mu$ and $\Sigma$ by given $\boldsymbol{\mu}_\theta(\mathbf{x}_t, \mathbf{y})$, $\boldsymbol{\Sigma}_\theta(\mathbf{x}_t, \mathbf{y})$, $y$ and $t$. And then we can sample $\mathbf{x}_{t-1}$ from,

$$\mathbf{x}_{t-1} \leftarrow \text{sampling from } \mathcal{N}(\mu + \Sigma F_\phi(\mathbf{x}_t, \mathbf{y}), \Sigma). \tag{13}$$

The second sampling algorithm, based on guided DDIM sampling, calculates a new $\hat{\epsilon}$ by given $\epsilon_\theta(\mathbf{x}_t, \mathbf{y})$. And then, we can sample $\mathbf{x}_{t-1}$ expressed by the following equation,

$$\hat{\epsilon} = \epsilon_\theta(\mathbf{x}_t, \mathbf{y})) - \sqrt{1-\bar{\alpha}_t}F_\phi(\mathbf{x}_t, \mathbf{y}), \tag{14}$$

$$\mathbf{x}_{t-1} = \sqrt{\bar{\alpha}_{t-1}}\Big(\frac{\mathbf{x}_t - \sqrt{1-\bar{\alpha}_t}\hat{\epsilon}}{\sqrt{\bar{\alpha}_t}}\Big) + \sqrt{1-\bar{\alpha}_{t-1}}\hat{\epsilon}. \tag{15}$$

## 4 EXPERIMENTS

We assess the effectiveness of our model in image super-resolution on human faces and natural images. Meanwhile, our method is not only suitable for image super-resolution but can also be

| Metric | PULSE | FSRGAN | Regression | SR3 | CG-DPM |
|--------|-------|--------|-----------|-----|--------|
| PSNR ↑ | 16.88 | 23.01 | 23.96 | 23.04 | 23.74 |
| SSIM ↑ | 0.44 | 0.62 | 0.69 | 0.65 | 0.71 |
| Consistency ↓ | 161.1 | 33.8 | 2.71 | 2.68 | 2.66 |

| Model | FID ↓ | IS ↑ | PNSR ↑ | SSIM ↑ |
|-------|-------|------|--------|--------|
| Reference | 1.9 | 240.8 | - | - |
| Regression | 15.2 | 121.1 | 27.9 | 0.801 |
| SR3 Saharia et al. (2022) | 5.2 | 180.1 | 26.4 | 0.762 |
| LDMs Rombach et al. (2022) | 4.3 | 174.9 | 24.7 | 0.710 |
| CG-DPM (Ours) | 4.8 | 188.7 | 27.1 | 0.783 |

Table 1: PSNR and SSIM on $16 \times 16 \rightarrow 128 \times 128$ face super-resolution. Consistency measures MSE ($\times 10^{-5}$) between the low-resolution inputs and the down-sampled super-resolution outputs.

Table 2: Quantitative evaluation with state-of-the-art methods for natural image super-resolution on the ImageNet validation set.

used to solve other generation tasks, e.g., medical image segmentation. Thus, we also conduct experiments on medical image segmentation tasks.

Our experiments consist of:

- Human face super-resolution at $16 \times 16 \rightarrow 128 \times 128$ and $64 \times 64 \rightarrow 512 \times 512$. The experiments are trained on FFHQ Karras et al. (2019) and evaluated on CelebA-HQ Karras et al. (2017).
- Natural image super-resolution at $64 \times 64 \rightarrow 256 \times 256$ and $128 \times 128 \rightarrow 512 \times 512$ on ImageNet.
- Medical image segmentation on the publicly available dataset, i.e., Automatic Cardiac Diagnosis Challenge (ACDC) Bernard et al. (2018).

**Datasets.** We follow the training strategies of the previous work SR3 Saharia et al. (2022), which trains face super-resolution models on Flickr-Faces-HQ (FFHQ) Karras et al. (2019) and evaluates on CelebA-HQ Karras et al. (2017). For natural images super-resolution, we train on ImageNet 1K Russakovsky et al. (2015) and use the dev split for evaluation. For training and testing, we utilize low-resolution images as conditional images involving our model. For ImageNet, we involve the class label for each image as the conditional information to our model. We use the largest central crop and then resize it to the target resolution using area resampling as high-resolution images. For medical image segmentation, we use a random split of 70 training cases, 10 validation cases, and 20 testing cases for the ACDC Bernard et al. (2018) dataset of 100 patients with the right ventricle (RV), myocardium (MYO), and left ventricle (LV) labels.

**Evaluation Metrics.** Following SR3 Saharia et al. (2022), we evaluate the model performance for face super-resolution via PSNR and SSIM. Besides PSNR and SSIM, we also employ FID and IS for natural image super-solution. For medical image segmentation, we evaluate performance via the average DSC.

## 4.1 COMPARISONS WITH STATE-OF-THE-ART METHODS

**Face Super-Resolution.** Figure 5(a) shows the qualitative results of the face super-resolution ($16 \times 16 \rightarrow 128 \times 128$). We compare our CG-DPM with recent methods such as SR3 Saharia et al. (2021), FSRGAN Chen et al. (2018), and PULSE Menon et al. (2020). The Regression baseline method baseline model uses the same architecture as SR3 but is trained with an MSE loss. Although the experiment adopts an $8 \times$ magnification factor, the inference images can still clearly see the detailed structure. Compared with SR3, the images inferred via our model are more like the reference images and more photo-realistic. Meanwhile, the PSNR and SSIM are better than SR3 in Table 1.

**Natural Image Super-Resolution.** In this part, the purpose is to prove the effectiveness of our guided network. We utilize a pre-trained model supplied by Dhariwal & Nichol (2021) that utilizes low-resolution images and class labels as conditional information, which is similar to SR3 but involves class labels. We build our guided network using the same architecture. But we utilize our proposal score-based loss function to train the guided network. Figure 5(b) and Figure 6(a) show the qualitative results of the natural super-resolution for $64 \times 64 \rightarrow 256 \times 256$ and $128 \times 128 \rightarrow 512 \times 512$ on the ImageNet dev set. The results demonstrate our model achieves better performance than SR3 since the generated images via our model contain more details structures and are more photo-realistic. Meanwhile, Table 3 illusrates the quantitative results, which show our CG-DPM outperforms SR3 for all metrics and outperforms Latent Diffusion Models (LDMs Rombach et al. (2022)) in IS, PNSR, and SSIM metrics, while LDMs has a better FID.

**Medical Image Segmentation.** In Table 4, we provide the quantitative experimental results on the ACDC dataset. Specifically, we compare our proposed CG-DPM with several leading convolution-based methods (i.e., R50-U-Net Ronneberger et al. (2015) and nnUNet Isensee et al. (2019)) and transformer-based methods (i.e., TransUNet Chen et al. (2021), SwinUNet Cao et al. (2021),

| Model | FID ↓ | IS ↑ | PNSR ↑ | SSIM ↑ |
|---|---|---|---|---|
| Reference | 1.9 | 240.8 | - | - |
| Regression | 15.2 | 121.1 | 27.9 | 0.801 |
| SR3 Saharia et al. (2022) | 5.2 | 180.1 | 26.4 | 0.762 |
| LDMs Rombach et al. (2022) | 4.3 | 174.9 | 24.7 | 0.710 |
| CG-DPM (Ours) | 4.8 | 188.7 | 27.1 | 0.783 |

| Method | Average ↑ | RV ↑ | Myo ↑ | LV ↑ |
|---|---|---|---|---|
| R50-U-Net Ronneberger et al. (2015) | 87.55 | 87.10 | 80.63 | 94.92 |
| R50-VIT-CUP Dosovitskiy et al. (2020) | 87.57 | 86.07 | 81.88 | 94.75 |
| UNETR Hatamizadeh et al. (2022) | 88.61 | 85.29 | 86.52 | 94.02 |
| TransUNet Chen et al. (2021) | 89.71 | 88.86 | 84.54 | 95.73 |
| nnUNet Isensee et al. (2019) | 91.20 | 89.30 | 89.09 | 95.20 |
| CG-DPM (Ours) | 89.34 | 87.63 | 85.26 | 95.13 |

Table 3: Quantitative evaluation with SOTA methods for natural image SR on the ImageNet val set.

Table 4: Quantitative evaluation with SOTA methods on the ACDC dataset (dice score in %).

and LeViT-UNet-384s Xu et al. (2021)). The results show that our proposed CG-DPM achieves competitive results compared to the traditional methods. Although the proposed CG-DPM does not achieve state-of-the-art results, our model performs the possibility that utilizes diffusion models to work on medical segmentation tasks. In Figure 6(b), we provide several qualitative results compared with several state-of-the-art methods, which demonstrate diffusion models have the ability to work on medical segmentation tasks.

## 4.2 ABLATION STUDY

We conduct extensive ablation studies on $16 \times 16 \rightarrow 128 \times 128$ face super-resolution to evaluate the proposed method. We have 5 baseline models (*i.e.*, B1, B2, B3, B4, and B5). The first four baseline models only have a single conditional diffusion model. **(i) B1** has an original U-Net. Conditional images connect with a series of stacked Down-Sampling layers, and then feed into the bottleneck. B1 is used to obtain semantic feature maps of conditional images and connect them with the bottleneck. **(ii) B2** has two of the same encoders for high-resolution images and conditional images, respectively, and then concatenates the two outputs to feed into the bottleneck. B2 is employed to obtain more semantic feature maps than B1. **(iii) B3** has the same structure as Figure 2(b). But the different scale feature maps are fed into the different scale levels in the encoder and the decoder. B3 is used to involve more conditional information in the whole U-Net. **(iv) B4** is our redesigned U-Net architecture without the guided network as shown in Figure 2(b). B4 is employed to involve different scales of conditional information in the encoder of the U-Net.

The results of the ablation study are shown in Table 5. Compared to B1, B2, and B3, the performance of B4 is the best in terms of both metrics, which demonstrates the effectiveness of our redesigned U-Net for image super-resolution tasks. Our redesigned U-Net equips each different scale level with independently stacked up-sampling or down-sampling blocks for the low-resolution images, which is effective to learn different scale information. Therefore, we utilize the same architecture for the guided network. **(v) B5** is our final model with a single conditional diffusion model and a guided network. B5 achieves the best results that demonstrate the effectiveness of our guided network, since it can guide toward the real target data points closer.

| # | Method | PSNR ↑ | SSIM ↑ |
|---|---|---|---|
| B1 | Stacked down-sampling for condition | 22.84 | 0.59 |
| B2 | Two same encoders for input & condition | 23.10 | 0.65 |
| B3 | Conditional features for encoder & decoder | 23.24 | 0.66 |
| B4 | CG-DPM without guided network | 23.51 | 0.68 |
| B5 | Our full model | **23.74** | **0.71** |

Table 5: Ablation studies on $16 \times 16 \rightarrow 128 \times 128$ face super-resolution.

## 5 CONCLUSION

We propose a novel conditional diffusion model with a score-based guided network, named the CG-DPM model. Particularly, we build a single diffusion model involving conditional images and redesign the architecture of the U-Net that involves conditional images into different scale levels in the encoder so that it can learn more different scale information about conditional images. In order for higher performance, we introduce a guided network trained via our proposal a score-based loss function to guide the single conditional diffusion model to generate high-resolution images toward the real target data points closer. Extensive experiments on face Karras et al. (2019; 2017) and natural Russakovsky et al. (2015) images super-resolution and medical image segmentation (i.e., ACDC Bernard et al. (2018)) datasets, demonstrate that our proposed CG-DPM model achieves strong performance compared with existing methods.

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
