# OpenReview forum: "Conditional Guided Diffusion Probabilistic Models for Image Super-Resolution"
_ICLR.cc/2024/Conference — ICLR 2024 Conference Withdrawn Submission_

### Official Review · Reviewer_AnkV · 2023-11-02

**Soundness:** 2 fair
**Presentation:** 2 fair
**Contribution:** 2 fair
**Rating:** 3
**Confidence:** 4

**Summary:**

This author proposes CG-DPM which adopts diffusion models for image SR task. Compared against SR3, CG-DPM involves conditional images in each different-scale level in the encoder and contains a guided network to predict the score-based likelihood function.

**Strengths:**

1. The method designed by the author seems quite reasonable, as it involves different scales of conditional image in the encoder module of U-Net. From the Table 5, it demonstrates the effectiveness of the redesigned U-Net.

**Weaknesses:**

1. The organization of the paper lacks logical coherence, making it very confusing.

   a) Regarding the method, in Section 3.2, the author introduces selected T, disparity between two cases in different steps, and sampling. However, it is not clear what the purpose of introducing these aspects is. For example, the author introduces two sampling methods, but what is the purpose of this introduction? Which sampling method is used in practice?

   b) In the score-based loss function part of Section 3.2, there are many places where the motivation is not well explained. Two more obvious instances are: "By the observation in Figure 4(c), xxxxxxxxxx. Therefore, xxxx", where the rationale for this design is not clearly explained; and "If we utilize the diffusion process, xxx for Equation 12", which is quite puzzling.

   c) The experimental section is also quite chaotic. Firstly, Table 2 and Table 3 are identical, and it's unclear why the task of medical segmentation is introduced. In addition, Table 2 shows the results of SR3 and LDMs for 64x64 to 256x256 resolution, but it is not clear which resolution the results for CG-DPM are based on.

2. The main experimental results in the paper are overly lacking.

   a) In the experimental section, the author mentioned that for both the human face super-resolution and natural image super-resolution tasks, there were two sets of experiments conducted under different resolutions. However, the author only presented the results for one set of resolutions. I suggest that the author supplement the paper with the results from both sets of experiments.

   b) Regarding the ablation study, the paper only provides numerical metrics without any qualitative analysis. In addition, there is no in-depth analysis to explain why the two designs proposed by the authors are effective. I would recommend the author to add more results and analysis.

   c) SR3 is a method proposed in 2021 and LDM is a method proposed in 2022. The author should compare it with more subsequent methods to more convincingly demonstrate the effectiveness of the proposed structure.

3. The writing of the paper needs to be improved, as there are many typos. Additionally, the overall readability of the paper is poor, making it difficult for readers to understand.

**Questions:**

Please refer to the weaknesses part.

---

### Official Review · Reviewer_iXKY · 2023-11-06

**Soundness:** 3 good
**Presentation:** 2 fair
**Contribution:** 2 fair
**Rating:** 5
**Confidence:** 4

**Summary:**

This paper proposes CG-DPM for image super-resolution. The authors redesign the U-Net architecture to involve conditional images and train a guided network to approximate the likelihood score function. Through extensive experiments, the proposed CG-DPM demonstrates strong results in image super-resolution tasks. The paper also highlights the potential application of the proposed method in other tasks, such as medical image segmentation.

**Strengths:**

- This paper redesigns the U-Net architecture that involves conditional images into each different scale level in the encoder.
- The idea of training a separate guided network to approximate the scored-based likelihood function is interesting.

**Weaknesses:**

- The evaluation is limited and weak.
    - In Table 1, PULSE (2020) and FSRGAN (2018) are outdated methods for face SR. The authors should include more advanced algorithms, *e.g,*. GPEN [1], GFPGAN[2], GLEAN[3], CodeFormer[4], for comparison.
    - For natural image SR, why not follow image SR conventions and train models on  DIV2K and evaluate on test set such as BSD100 or Urban? In addition, more SOTA methods of SR should be included.
    - Qualitative results are not sufficient to reveal the effectiveness of the proposed method. Similar to the above issues, more SOTA should be included for comparison.

Given the weak evaluation, the reviewer cannot properly assess the effectiveness of the proposed model.

**Questions:**

See weaknesses

---

### Official Review · Reviewer_dDwK · 2023-11-08

**Soundness:** 2 fair
**Presentation:** 2 fair
**Contribution:** 2 fair
**Rating:** 3
**Confidence:** 5

**Summary:**

This paper proposes a novel conditional guided diffusion model (CG-DPM) for image super-resolution. Combines a conditional diffusion model and a separate score-based guided network. The conditional diffusion model uses a redesigned U-Net architecture that incorporates the low-resolution image at multiple scales in the encoder. This allows it to better learn conditional information. The guided network is trained with a novel score-based loss to predict the score (gradient of the log-likelihood) and guide the diffusion model closer to the target high-resolution image.

**Strengths:**

Novel score-based loss for training guided network is an interesting idea to help diffusion modeling for pixel-level tasks.

**Weaknesses:**

First, the writing of this paper needs substantial improvement. The overall writing is wordy. There are a lot of typos. And the summary and expression of its methods are not in place. Many mathematical symbols are used without explanation. I don't think this paper is ready for publication just in terms of presentation.

Secondly, I think innovation in network architecture is unimportant. At least for the topic discussed in this paper. Modifying the network, especially in such an intuitive way, brings very limited insights. Its ablation experiments are also mainly about network architecture. I am very interested in the second part of the method, but the author seems to (1) not explain and demonstrate this method in depth; (2) does not discuss the actual role and necessity of this method. This is disappointing. In-depth ablation studies and interpretable visualization and interpretation of this part are required.

The comparison of results is also incomplete, as some recent methods were not included. Figure 1 is too small to see the details of the image. The paper added experiments on segmentation, but this did not help this paper. Instead, it took up space that should have been used to explain the method in depth.

The authors are advised to rethink parts of their approach other than network architecture and substantially improve the presentation of the article.

**Questions:**

see Weakness